# The Effect of Velocity Loss on Strength Development and Related Training Efficiency: A Dose–Response Meta–Analysis

**DOI:** 10.3390/healthcare11030337

**Published:** 2023-01-23

**Authors:** Xing Zhang, Siyuan Feng, Hansen Li

**Affiliations:** 1Key Laboratory of Physical Fitness Evaluation and Motor Function Monitoring, Institute of Sports Science, College of Physical Education, Southwest University, Chongqing 400715, China; 2Laboratory of Genetics, University of Wisconsin-Madison, Madison, WI 53706, USA

**Keywords:** velocity–based training, resistance training, one–repetition maximum, maximum strength, training efficiency

## Abstract

The velocity loss method is often used in velocity–based training (VBT) to dynamically regulate training loads. However, the effects of velocity loss on maximum strength development and training efficiency are still unclear. Therefore, we conducted a dose–response meta–analysis aiming to fill this research gap. A systematic literature search was performed to identify studies on VBT with the velocity loss method via PubMed, Web of Science, Embase, EBSCO, and Cochrane. Controlled trials that compared the effects of different velocity losses on maximum strength were considered. One–repetition maximum (1RM) gain and 1RM gain per repetition were the selected outcomes to indicate the maximum strength development and its training efficiency. Eventually, nine studies with a total of 336 trained males (training experience/history ≥ 1 year) were included for analysis. We found a non–linear dose–response relationship (reverse U–shaped) between velocity loss and 1RM gain (*p_dose–response relationship_* < 0.05, *p_non–linear relationship_* < 0.05). Additionally, a negative linear dose–response relationship was observed between velocity loss and 1RM gain per repetition (*p_dose–response relationship_* < 0.05, *p_non–linear relationship_* = 0.23). Based on our findings, a velocity loss between 20 and 30% may be beneficial for maximum strength development, and a lower velocity loss may be more efficient for developing and maintaining maximum strength. Future research is warranted to focus on female athletes and the interaction of other parameters.

## 1. Introduction

Maximum strength development is critical for improving force–time characteristics and thereby, various athletic performances [1], including but not limited to jumping [2,3], sprinting [4], changing direction [5], and even specific sports skills [1]. Furthermore, greater strength is beneficial for the structural strength of ligaments, tendons, tendons/ligaments to bone junctions, joint cartilage, and connective tissue sheaths within the muscle, thereby reducing the risks in sports [6]. For these reasons, many efforts have been made to find effective training methods for developing maximum strength.

In the past decades, traditional resistance training (RT) methods have been viewed as a “gold criterion” in prescribing training loads for maximizing muscular strength [7]. However, traditional RT loads are usually designed based on individuals’ 1RM (one–repetition maximum) before starting an RT session [8]. Such predesigned training loads rarely consider athletes’ daily fluctuation in training state or performance [9,10], which may lead to inappropriate training loads, lower training benefits, and even degeneration or injuries [10]. In this context, a series of regulable and flexible RT methods, known as autoregulation methods, were invented to avoid this limitation of traditional RT.

Velocity loss (VL) is a critical index/parameter of velocity–based training (VBT) and is often applied to determine the number of repetitions in a training set [9]. Specifically, a training set will be called to stop when the velocity loss exceeds a target value. To date, some scholars have focused on the role of velocity loss methods in developing athletic performances [11,12,13,14], but the optimal velocity loss for maximizing strength adaption is still inconclusive.

In practice, competitive athletes usually have complex training arrangements, especially in competitive seasons. They may have fewer RT sessions and, in turn, a lower maximum strength [15,16]. Therefore, how to maintain and enhance maximum strength with a lower RT volume becomes a valuable research question. Several previous studies on VBT have implied that a lower velocity loss and its higher counterpart may be or similarly effective in maximum strength development, but the lower velocity loss may result in a lower training volume under the same relative intensity [17,18]. In other words, one repetition in a training intervention may lead to a greater gain of maximum strength on average under a lower velocity setting. This topic of training efficiency, however, has only been studied concerning traditional RT [19], while the relationship between velocity loss and training efficiency in VBT is understudied yet. Therefore, we conducted this dose–response meta–analysis aiming to:

1. Examine the effect of velocity loss on maximum strength development;

2. Examine the effect of velocity loss on the efficiency of maximum strength development.

## 2. Materials and Methods

This review was conducted following the recommendations for the preferred reporting items for systematic reviews and meta–analyses (PRISMA) [20]. However, due to the nature of the research question, outcome, and paper organization, we could not register our review in PROSPERO or other alternative databases [21].

### 2.1. Eligibility Criteria

Aiming to serve athletic training, studies on sportspeople who have at least a year of RT experience/history were considered. Studies that selected squat, bench press, and deadlift as the major training event were considered [22]. To investigate the long–term effect, interventions that lasted for at least four weeks were considered [23]. Guided by previous reviews, we used the one repetition maximum (1RM) for indicating maximum strength [24]. 

Our eligibility criteria obeyed the PICOS principle as follows [25]:

–P (population): people who have RT experience/history for at least one year;

–I (intervention): at least four weeks of VBT intervention that selected squat, bench press, and deadlift as their training events;

–C (comparison): compared the effect of VBT with different velocity losses on maximum strength;

–O (outcomes): 1RM was measured in the training events;

–S (study design): controlled trials based on pre–post designs that evaluated the effect of different velocity loss methods on maximum strength.

### 2.2. Information Sources and Search Strategy

A systematic literature search was conducted through the following English electronic databases: PubMed, Web of Science, Embase, EBSCO, and Cochrane. The time range was between the inception of each database and January 12, 2023. The detailed search strategy is shown in Table 1.

### 2.3. Study Selection

Two authors (XZ and HSL) independently selected relevant studies based on their titles, abstracts, and full texts. Any discrepancies were resolved by discussion or judgments from another author (SYF) [26].

### 2.4. Data Collection Process and Data Items

Two authors (XZ and HSL) independently extracted data from the included articles, including title, publication year, author name, study design, participant profile, sample size, intervention, measurement, and outcomes. Any discrepancies were resolved by discussion or judgments from a third author (SYF). Since 1RM is the most commonly used measure for evaluating maximum strength performance [8,27], it was selected as the outcome in the current study. 

### 2.5. Risk of Bias Assessment

The risk of bias in the included studies was assessed using the PEDro scale (physiotherapy evidence–based database). According to the previous study, the PEDro scale was evaluated to have a high reliability and validity [28,29]. Items 2 to 11 were used to calculate the PEDro score. The methodology criteria were scored as: “Yes” (one point), “No” (zero points), or “Don’t know” (zero points). The total PEDro score indicates the overall quality of each study (9–10 = excellent; 6–8 good; 4–5 = fair; and <4 = poor). Two authors (XZ and HSL) independently assessed the potential risk of bias. Any discrepancies were resolved by discussion or judgments from another author (SYF). 

### 2.6. Statistical Analysis

#### 2.6.1. Variable Calculation

The maximum strength development was represented by the 1RM gain throughout the intervention, which was calculated as the mean change from the baseline to the final test (post–pre interventions). The change in SD of the 1RM was calculated according to the Cochrane Handbook for Systematic Reviews of Interventions (Section 6.5.2.8) [30]. 

As the authors of the included studies did not provide correlations upon our request, we followed the method of others and assumed a conservative correlation coefficient of 0.5 for calculating [31]. This formula is as follows: $$SD_{change}=\surd (SD_{baseline}^{2}+SD_{final}^{2}-\left(2\times 0.5\times SD_{baseline}\times SD_{final}\right))\;$$
where SD indicates the standard deviation, and baseline and final indicate the outcome measured before and after an intervention.

The 1RM gain per repetition was used to assess the efficiency of different velocity loss methods and was calculated as the mean change (post–pre) in the maximum strength results divided by the total repetition throughout the intervention. 

#### 2.6.2. Dose–Response Meta–Analysis

The dose–response meta–analysis was carried out to examine the “velocity loss–1RM gain” and the “velocity loss–1RM gain per repetition” relationships. The robust error meta–regression (REMR) model was used for analyzing dose–response relationships [32]. This is a one–stage method that considers all included studies as a whole and treats each study as a cluster in order to validate the fitting of the dose–response curve. In addition, in order to consider both linearity and non–linearity in one model, we used the restricted cubic spline (RCS) function to fit the dose–response trend. The random knots were used in the RCS function [33]. Three random knots divided the data into four pieces, and the dose–response curve was fitted within each piece and was further smoothed at the knots. The first and last pieces were restricted as linear, while for the second and third pieces, a cubic spline was fitted. When the cubic and quadratic terms in the function are equal to zero, the RCS automatically degrades into a simple linear function. The Wald test was used to test the probability that the cubic and quadratic terms equal zero [34]. Egger’s test was conducted to evaluate the publication bias [35]. All analyses were performed using STATA 14.0 (StataCorp LLC, Texas, US), and a *p*–value < 0.05 was considered statistically significant [36,37]. 

## 3. Results

### 3.1. Study Selection

The systematic literature search and study selection processes are outlined in Figure 1. A total of 3438 studies were identified in the search. After that, 3423 were excluded due to duplication, title, and abstract. Six studies were excluded for inappropriate controls or missing outcomes. Eventually, nine studies were included in the current study.

### 3.2. Study Characteristics

Nine studies and 336 subjects were included in the current study (Table 2). The studies were published between 2017 to 2022. All subjects were trained males (at least one year of RT history or from professional clubs). Most studies (*n* = 6) deployed an 8–week VBT intervention, and three other studies respectively performed a 5–week, a 6–week, and a 7–week VBT intervention, respectively. Most studies (*n =* 6) selected squat as the training events, and three studies used bench presses. All studies tested the 1RM as the primary outcome. All the included studies were controlled for relative intensity, the number of sets, and recovery time/interval. The range of velocity was between 0% and 50%. 

### 3.3. Risk of Bias

Seven studies were assessed as good and two as fair quality according to the PEDro scale (Table 3). All studies were short in blinding, including subject, therapist, and assessor blinding.

### 3.4. Dose–Response Meta–Analysis 

#### 3.4.1. Maximum Strength Gain

Nine studies were included in the dose–response meta–analysis concerning the effect of velocity loss on maximum strength development. Figure 2 and Table 4 demonstrate a non–linear dose–response relationship (reverse U–shaped) between velocity loss and 1RM gain (*p_dose–response relationship_* < 0.05, *p_non–linear relationship_* < 0.05). The 1RM gain increased and then decreased with the velocity loss, and the greatest values were observed between VL20% and 30%.

#### 3.4.2. Maximum Strength Gain Per Repetition

Nine studies were included in the dose–response meta–analysis concerning the effect of velocity loss on the efficiency of maximum strength development. Figure 3 and Table 5 demonstrate a linear dose–response relationship between velocity loss and 1RM gain per repetition (*p_dose–response relationship_* < 0.05, *p_non–linear relationship_* = 0.23). The efficiency consistently decreased with velocity loss, and the highest efficiency was observed at VL0%.

### 3.5. Publication Bias Assessment

The Egger’s test revealed a significant publication bias in the results of 1RM gain (*p* < 0.05) and 1RM gain per repetition (*p* < 0.05), indicating a negative impact on our estimation.

## 4. Discussion

This study aimed to explore the effects of velocity loss of VBT on maximum strength development and training efficiency. The results generally showed an optimal range of velocity loss (20–30%) for maximum strength development. Moreover, we found that training efficiency broadly decreased with velocity loss, which means that a lower velocity loss may help to reduce the required training volume of RT sessions. These findings may offer some indications for the users and developers of VBT. 

### 4.1. The Relationship between Velocity Loss on Maximum Strength Development

The optimal velocity loss range for maximizing strength adaptations has been widely discussed by strength and conditioning professionals [17,38,39,42]. For instance, a narrative review by Włodarczyk, et al. [45] demonstrated that a velocity loss between 10% and 20% could be helpful for maximum strength development among elite athletes. However, this finding is not supported by quantitative methods and may not address the existing controversies. A later study by Hernández-Belmonte and Pallarés [46] further provided a meta–analysis and compared the effects of the low–moderate (VL ≤ 25%) and the moderate–high velocity loss groups (VL > 25%) on maximum strength growth. Their observed differences were not statistically significant, which indicates a similar role of low–moderate and moderate–high velocity loss in strength development. However, the pairwise comparison that roughly divided the velocity range into two categories could not contribute to uncovering the relationship between velocity loss and training outcomes, particularly given the fact that some emerging studies have used other velocity loss settings. Our study, in comparison, included a wider velocity loss range and evaluated the dose–response relationship. We observed a reverse U–shaped relationship between velocity loss and 1RM gain. According to these results, the range between VL20% and VL30% might be most helpful for maximum strength development. 

In physiological theories, changes in any athletic performance can be explained by the physiological adaptations induced by training stimulations [47,48]. Pareja-Blanco, et al. [44] found that VBT with a high velocity loss increased the cross–sectional area of slow–twitch fibers, suggesting a negative impact on maximum strength development. A recent review suggests that the velocity loss of VBT is negatively associated with the IIX (MHC–IIX) percentage and is positively associated with the myosin heavy chain I (MHC–I) percentage [9], which may explain the selective hypertrophy of skeletal muscles in VBT. In other words, a high velocity loss is beneficial for endurance–related performance instead of for maximum strength. These theories appear to go against our findings. One potential reason is that a too–low velocity loss may lead to an insufficient training volume. Specifically, our included studies controlled their training sets, so their velocity loss might largely influence their repetitions and therefore the training volume per session. For example, two included studies deployed six repetitions a week (1 rep × 3 sets × 2 sessions) for their VL0% groups. This “tiny” training volume may lead to insufficient stimulation for muscle fibers’ growth and maximum strength development [49].

### 4.2. The Relationship between Velocity Loss and the Efficiency of Maximum Strength Development

Previous experimental studies and reviews have suggested the high efficiency of low velocity loss in promoting strength adaptions [17,18,46]. To our knowledge, no quantitative evidence has been found in relevant studies. This research gap has motivated this dose–response meta–analysis to examine the relationship between them. Our results revealed a consistently negative association between velocity loss and the 1RM gain per repetition. This finding indicates that a lower velocity loss may develop maximum strength with a lower training volume, which supports the efficiency of low velocity loss methods, although a low velocity loss is not the best option for maximizing strength adaption.

As mentioned in the Introduction, maintaining maximum strength is important for competitive athletes during a competitive season. Unfortunately, the truth is that the decline in strength is a highly common phenomenon in competitive seasons, especially among rugby [16], football [50], basketball [15], and baseball [51] players. A major reason for this phenomenon is the reduction in RT sessions resulting from frequent competitions [16,52,53]. Based on our findings, we recommend using VBT with a low velocity loss as an RT prescription during a competitive season because it may require a lower training volume and may therefore save energy and time for other training arrangements.

### 4.3. Limitations

Several limitations should be noted. First, only male samples were involved in the relevant interventions, so the findings of the current study may not be generalized to females. Second, samples were insufficient for assessing some velocity loss settings. For example, only one included study used VL5%, and no included study used VL35%. Thus, re–examination is needed when more relevant studies emerge. Third, we could not perform extra dose–response meta–analyses for squat or bench press due to insufficient samples. Hence, lower– and upper–body differences are unclear. Fourth, the current study only focused on maximum strength. Future research may check the effect of velocity loss on other indices, such as the maximal number of repetitions (MNR), countermovement jump (CMJ), and sprint time. Finally, and the most importantly, we only focused on the relationship between velocity loss and the outcomes of interest, while the impacts of other parameters (e.g., number of sets) were not controlled, and we could not reveal the potential interaction of different training parameters. These are the inherent limitations of such a research approach [54,55]. Thus, our findings must be applied to practice with caution.

## 5. Conclusions

In the current study, we examined the effects of velocity loss on maximum strength development and relevant training efficiency. Our results revealed a reverse U–shaped relationship between velocity loss and maximum strength gain, and the velocity loss range between 20 and 30% might most effectively contribute to maximum strength development among trained individuals. Meanwhile, a negative relationship was observed between velocity loss and the efficiency of maximum strength development, implying a direction to reduce total training volume and save energy for other training in some special scenarios. These findings may offer quantitative evidence to reinforce some previous studies and address some controversies, but they should also be carefully applied to practice given the several mentioned limitations. 

## Figures and Tables

**Figure 1 healthcare-11-00337-f001:**
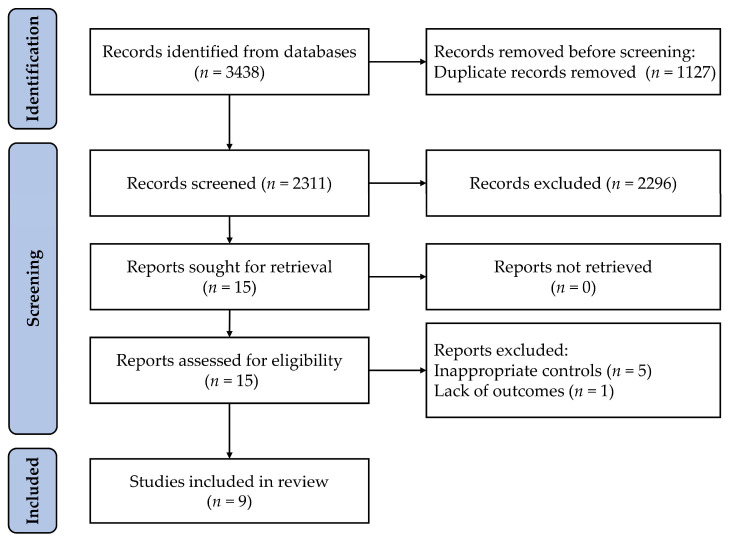
Flow diagram of the screening and selection of studies.

**Figure 2 healthcare-11-00337-f002:**
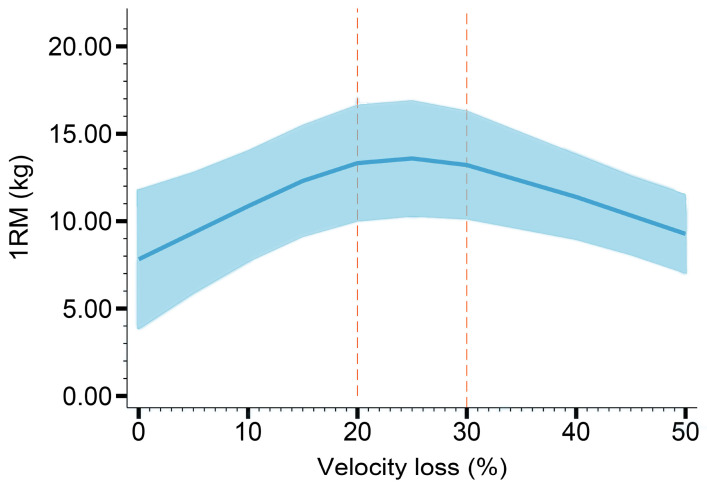
The dose–response curve of 1RM gain.

**Figure 3 healthcare-11-00337-f003:**
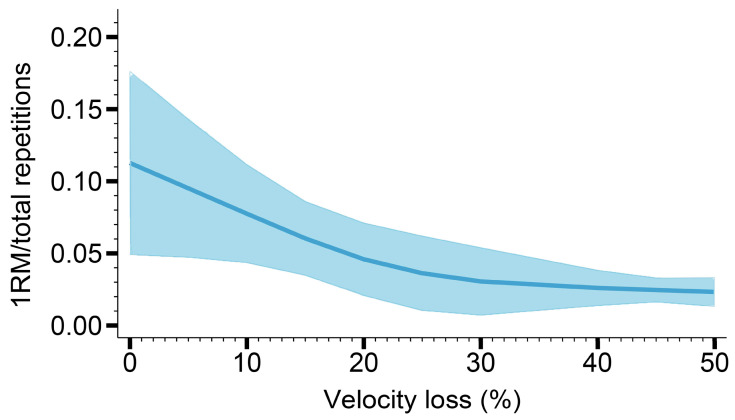
The dose–response curve of the maximum strength gain per repetition.

**Table 1 healthcare-11-00337-t001:** Searching strategy for study inclusion.

Steps	Searching Command	Field
#1	Velocity–based training OR VBT OR velocity–based resistance training OR VBRT OR velocity loss OR VL	Title or abstract
#2	Maximum strength OR one–repetition maximum OR 1RM OR strength performance	Title or abstract
#3	#1 AND #2	

**Table 2 healthcare-11-00337-t002:** The characteristics of the included studies.

Authors	Velocity Loss (Sample)	Gender	Age (Years)	Experience (Years)	Intervention (Weeks)	Training Design	Total (Repetitions)	Outcome
Galiano, et al. [38]	VL5 (*n* = 15) VL20 (*n* = 13)	Male	23.0 ± 3.2	At least 1.5 years	7 weeks	Event: squats Intensity: 50% 1RM Set: 3 sets Recover time: 3 min Frequency: twice a week	VL5 (156.9) VL20 (480.5)	1RM
Rodríguez-Rosell, et al. [39]	VL10 (*n* = 11) VL30 (*n* = 11) VL45 (*n* = 11)	Male	22.8 ± 3.9	At least 1 year	8 weeks	Event: squats Intensity: 55–70% 1RM Set: 3 sets Recover time: 4 min Frequency: twice a week	VL10 (180.8) VL30 (347.9) VL45 (501.1)	1RM
Rodríguez-Rosell, et al. [40]	VL10 (*n* = 12) VL30 (*n* = 13)	Male	≈22.49	At least 1 year	8 weeks	Event: squats Intensity: 70–85% 1RM Set: 3 sets Recover time: 4 min Frequency: twice a week	VL10 (109.6) VL30 (228.0)	1RM
Rodiles-Guerrero, et al. [41]	VL10 (*n* = 15) VL30 (*n* = 15) VL50 (*n* = 15)	Male	23.0 ± 2.0	At least 1 year	5 weeks	Event: bench presses Intensity: 65–85% 1RM Set: 4 sets Recover time: 3 min Frequency: three times a week	VL10 (211.1) VL30 (398.1) VL50 (444.4)	1RM
Rodiles-Guerrero, et al. [42]	VL0 (*n* = 12) VL15 (*n* = 13) VL25 (*n* = 13) VL50 (*n* = 12)	Male	23.3 ± 3.3	At least 1.5 years	8 weeks	Event: bench presses Intensity: 55–70% 1RM Set: 3 sets Recover time: 4 min Frequency: twice a week	VL0 (48) VL15 (189.4) VL25 (310.2) VL50 (490.9)	1RM
Pareja-Blanco, et al. [43]	VL15 (*n* = 8) VL30 (*n* = 8)	Male	23.8 ± 3.4	Professional soccer club	6 weeks	Event: squats Intensity: 50–70% 1RM Set: 2–3 sets Recover time: 4 min Frequency: three times a week	VL15 (251.2) VL30 (414.6)	1RM
Pareja-Blanco, et al. [44]	VL20 (*n* = 12) VL40 (*n* = 10)	Male	22.7 ± 1.9	At least 1.5 years	8 weeks	Event: squats Intensity: 69–85% 1RM Set: 3 sets Recover time: 4 min Frequency: twice a week	VL20 (185.9) VL40 (310.5)	1RM
Pareja-Blanco, et al. [18]	VL0 (*n* = 14) VL10 (*n* = 14) VL20 (*n* = 13) VL40 (*n* = 14)	Male	24.1 ± 4.3	At least 1.5 years	8 weeks	Event: squats Intensity: 70–85% 1RM Set: 3 sets Recover time: 4 min Frequency: twice a week	VL0 (48.0) VL10 (143.6) VL20 (168.5) VL40 (305.6)	1RM
Pareja-Blanco, et al. [17]	VL0 (*n* = 15) VL15 (*n* = 16) VL25 (*n* = 15) VL50 (*n* = 16)	Male	24.1 ± 4.3	At least 1.5 years	8 weeks	Event: bench presses Intensity: 70–85% 1RM Set: 3 sets Recover time: 4 min Frequency: twice a week	VL0 (48.0) VL15 (136.6) VL25 (191.1) VL50 (316.4)	1RM

**Note:** VL, velocity loss; 1RM, one–repetition maximum; The same raining variables, if the main training variable is the same within the study, including the relative intensity, number of sets, and recover time.

**Table 3 healthcare-11-00337-t003:** The risk of bias assessment for the included studies.

Studies	PEDro Item											Assessment
	1	2	3	4	5	6	7	8	9	10	11	
Galiano, et al. [38]	Yes	1	–	1	–	–	–	1	1	1	1	good
Rodríguez-Rosell, et al. [39]	Yes	1	–	1	–	–	–	1	1	1	1	good
Rodríguez-Rosell, et al. [40]	Yes	1	–	1	–	–	–	1	1	1	1	good
Rodiles-Guerrero, et al. [41]	Yes	1	–		–	–	–	1	1	1	1	fair
Rodiles-Guerrero, et al. [42]	Yes	1	–	1	–	–	–	1	1	1	1	good
Pareja-Blanco, et al. [43]	Yes	1	–	1	–	–	–		1	1	1	fair
Pareja-Blanco, et al. [44]	Yes	1	–	1	–	–	–	1	1	1	1	good
Pareja-Blanco, et al. [18]	Yes	1	–	1	–	–	–	1	1	1	1	good
Pareja-Blanco, et al. [17]	Yes	1	–	1	–	–	–	1	1	1	1	good

**Note:** Item 1. Eligibility criteria were specified. 2. Subjects were randomly allocated to groups (in a crossover study, subjects were randomly allocated an order in which treatments were received). 3. Allocation was concealed. 4. The groups were similar at baseline regarding the most important prognostic indicators. 5. There was blinding of all subjects. 6. There was blinding of all therapists who administered the therapy. 7. There was blinding of all assessors who measured at least one key outcome. 8. Measures of at least one key outcome were obtained from more than 85% of the subjects initially allocated to groups. 9. All subjects for whom outcome measures were available received the treatment or control condition as allocated, or where this was not the case, data for at least one key outcome was analyzed by “intention to treat.” 10. The results of between–group statistical comparisons were reported for at least one key outcome. 11. This study provides both point measures and measures of variability for at least one key outcome.

**Table 4 healthcare-11-00337-t004:** The dose–specific effects of velocity loss on maximum strength gain and the model fit.

Velocity	ES	95%CI
VL0	7.82	3.84 to 11.80
VL5	9.34	5.88 to 12.79
VL10	10.86	7.68 to 14.03
VL15	12.30	9.12 to 15.49
VL20	13.32	10.00 to 16.64
VL25	13.59	10.28 to 16.89
VL30	13.21	10.12 to 16.30
VL40	11.38	8.95 to 13.82
VL45	10.33	8.07 to 12.58
VL50	9.27	7.02 to 11.53
**Wald test**		
Dose–response relationship		*p* < 0.05
Non–linear relationship		*p* < 0.01

**Note:** ES, estimate; CI, confidence interval.

**Table 5 healthcare-11-00337-t005:** The dose–specific effects of velocity loss on the maximum strength gain per repetition and the model fit.

Velocity	ES	95%CI
VL0	0.11	0.05 to 0.18
VL5	0.10	0.05 to 0.14
VL10	0.08	0.04 to 0.11
VL15	0.06	0.03 to 0.09
VL20	0.05	0.02 to 0.07
VL25	0.04	0.01 to 0.06
VL30	0.03	0.01 to 0.05
VL40	0.03	0.01 to 0.04
VL45	0.02	0.02 to 0.03
VL50	0.02	0.01 to 0.03
**Wald test**		
Dose–response relationship		*p* < 0.05
Non–linear relationship		*p* = 0.23

**Note:** ES, estimate; CI, confidence interval.

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
