# Peer review of "The Effect of Velocity Loss on Strength Development and Related Training Efficiency: A Dose–Response Meta–Analysis"

_healthcare, 2023, doi:10.3390/healthcare11030337_

Round 1
Reviewer 1 Report
The manuscript “The effect of velocity loss on maximum strength development and the efficiency of maximum strength development: a systematic review and dose-response meta-analysis” aimed to investigate the impacts of different magnitudes of velocity loss during velocity-based training in developing maximum strength as well as to determine the efficiency of different levels of velocity loss on the same outcome. I commend the authors for their work in conducting the experiment and putting the manuscript together. General comments and specific points and sections are provided below:
General comments
Presentation:
- The English writing is fine and does not require professional revisions
Title:
- The title seems odd to me. When I first read it, I did not get the picture of the study, mostly due to the repetition of “maximum strength development”. Please consider revising for clarity.
Abstract:
- The abstract is concise and well-written. It provides the main findings of the study successfully.
Introduction:
- The introduction is well-written and provides a good overview of the literature while also conducting the rationale for the study.
Materials and Method:
- The procedures for study selection and data extraction followed the PRISMA guidelines and the PICOS principle is clearly stated. The quality assessment is adequate and based off previous reliable studies and methods.
- This reviewer does not feel qualified to judge the statistical procedures for meta-analyses.
Results:
- The qualitative description of the studies is nice and organized.
- Quality assessment is clear and coherent.
- The quantitative results are presented very informatively in both text and figures/tables
Discussion:
- The discussion is short as it usually is for meta-analyses. However, it addresses the main findings and contextualizes them with the literature successfully.
- The limitations are clearly presented, and I commend the authors for this.
- I do not believe this is a limitation of the study, but it should be discussed that the risk of bias for this review is considerable since there are few research groups investigating this phenomenon (two, if I am not wrong).
Conclusion:
- The conclusion is concise and based off the obtained results, as should be.
Please find specific comments detailed below:
L152: Except or besides? I reckon all the studies did control for velocity loss.
Author Response
We thank the reviewer for assessing the work and the encouraging words. Please check those revisions (in blue) in the revised manuscript.
Reviewer 2 Report
Dear authors
You described a well-defined research gap, aiming to test the effect of velocity loss on maximum strength development and the efficiency of maximum strength development. The quality of the study elaboration and results are clearly described in the text. Generally, the text should be improved in the sense of coherence of phrases. At some points, I perceived possible improvements, so, I gave my observations, which are marked in the pdf document. No further remarks.
Best Regards

Author Response
We thank the reviewer for his/her time into this work. We have improved the paper according to your suggestion. Please check the revisions in blue in the revised manuscript.
Reviewer 3 Report
The authors of this study try to present evidence of the impact of velocity loss on maximum strength development 2 and the efficiency of maximum strength development. The current study conducted by the authors provides the most comprehensive analysis data. However, I still have some concerns regarding the Methodological quality of this study. Below you can see my detailed comments regarding this study:
1. Has this systematic review been registered in PROSPERO? If yes, please provide the registration number. If not, please explain the reasons.
2. The last update of the systematic search was June 23, 2022, why this date has not been updated?
3. Statistical analysis needs to present more details about ES measurement, confidence interval assessment, heterogeneity assessment, and all statistical analysis related to the present study.
4. Overall, the discussion looks good, but more physiological mechanisms related to skeletal muscle hypertrophy need to be presented. Please use the following reference for providing more information regarding that concern: 10.1002/jcsm.13043.
5. Literature screening and data extraction need reference citation. The authors can use the following reference: 10.1093/ptj/pzab144.
Author Response
We thank the reviewer for reviewing our work and offering helpful suggestions. Please check the responses and revisions below.

Round 2
Reviewer 3 Report
Congratulations to the authors for the great work done in the revised version.
Author Response
Dear Reviewer,
Thank you for your help and support!
Best regards,
XZ